# The Chinese version of the Maltreatment and Abuse Chronology of Exposure (MACE) scale: Psychometric properties in a sample of young adults

**Yuanyuan Chen[1], Zhen Wang[2], Xiaoyu Zheng[3], Zhiyin Wu[3], Jianjun Zhu**[1,4]*

**1** School of Education, Guangzhou University, Guangzhou, China, **2** Department of Psychology, Guangzhou Xinhua University, Dongguan, China, **3** School of Education, Hangzhou Normal University, Hangzhou, China, **4** Department of Psychology, Guangzhou University, Guangzhou, China

* jianjunzhu722@gmail.com

## Abstract

There are several effective self-report instruments used by Chinese researchers to retrospectively assess exposure to childhood maltreatment. However, these measures do not assess the timing of exposure, restricting efforts to identify periods of development when childhood maltreatment maximally increases vulnerability to psychopathology and health outcomes. In the current study we created a Chinese version of the Maltreatment and Abuse Chronology of Exposure (MACE) scale, which assesses multiplicity (number of types of maltreatment experienced) and severity of maltreatment as well as when it occurred during childhood and adolescence. Rasch modeling was used for scale development in a sample of 812 undergraduate students. Item reduction analysis of the original 75 items produced a 58-item Chinese version with ten subdimensions. The new scale showed good three-week test-retest reliability, and good convergent validity with the Childhood Trauma Questionnaire (CTQ) and the revised Adverse Childhood Experiences Questionnaire (ACEQ-R). Variance decomposition analyses found that compared to the CTQ and ACE, the MACE Severity and Multiplicity scores explained higher variance in self-reported depression and anxiety symptom ratings on the Depression Anxiety Stress Scales (DASS). The results of the present study confirmed that the Chinese version of the MACE has sound psychometric properties in the Chinese cultural context. This new instrument will be a valuable tool for Chinese researchers, psychiatrists and psychologists to ascertain the type and timing of exposure to maltreatment.

## Introduction

Childhood maltreatment is a relatively common adverse experience [1], which has been widely implicated as a risk factor for adult psychiatric disorders, such as major depressive disorder [2, 3], anxiety disorder [4, 5], post-traumatic stress disorder [6], and substance abuse [7]. Further,

**Data Availability Statement:** Data could be found in the Supporting information.

**Funding:** This research was supported by the grant from the National Natural Science Foundation of

China (32000755) to Jianjun Zhu. The funder had no role in study design, data collection and analysis, decision to publish, or preparation of the manuscript.

**Competing interests:** The authors have declared that no competing interests exist.

childhood maltreatment is associated with earlier onset of mood disorders, as well as more recurrent episodes and poorer treatment response [8]. The population-attributional risk fraction explained by childhood maltreatment and household dysfunction for a wide range of disorders is between 30% and 70% [5, 9–11].

There are numerous questionnaire measures that assess adults' retrospective reports of the severity and multiplicity of exposure to childhood maltreatment during the first 18 years of life. The Childhood Trauma Questionnaire (CTQ) [12, 13] and the Adverse Childhood Experience Scale (ACE) [14] are currently the most commonly used instruments of this type. The 28-item CTQ assesses the severity of exposure to five types of maltreatment (i.e., physical abuse, emotional abuse, sexual abuse, emotional neglect and physical neglect), and the 18-item ACE measures the multiplicity of exposure to three categories of childhood abuse (i.e., emotional, physical, and sexual abuse) as well as five categories of household dysfunction (i.e., exposure to substance abuse, mental illness, violent treatment of mother or stepmother, incarceration for criminal behavior, and parental separation, divorce or death). Both of CTQ and ACE have been shown to have good reliability and validity [12–14]. Finkelhor et al. (2015) added items to the ACE to assess exposure to the adverse experiences of peer victimization, peer isolation/rejection, and community violence, constituting a revised Adverse Childhood Experiences Questionnaire (ACEQ-R) with satisfactory reliability and validity. Some researcher have introduced CTQ, ACE and ACEQ-R into China [15, 16] and demonstrated that these measures are good tools for evaluating child adversity of Chinese people.

Although the CTQ and the ACE are popular instruments for measuring child maltreatment, two critical limitations need to be mentioned. First, the CTQ and ACE do not assess some important types of maltreatment, such as witnessing violence between parents or towards siblings, although these have been shown to be strong risk factors for psychiatric symptoms [17–19]. Second, neither instrument collects information on the timing of exposures to maltreatment over the course of childhood and adolescence. This consideration is critically important as there may be periods of development when exposure to specific types of maltreatment is maximally associated with vulnerability for psychopathology and alterations in the structure and function of stress-susceptible brain regions [9]. Ascertaining whether the consequences of maltreatment vary as a function of when the exposure occurred may have important practical application. The information could guide research on when interventions are most likely to be effective, and prompt innovative approaches for preventing the onset and development of psychopathology [20].

Teicher and Parigger (2015) developed the Maltreatment and Abuse Chronology of Exposure (MACE) scale to address the limitations of earlier retrospective questionnaire measures of child maltreatment. First, the MACE assesses whether or not there was exposure, and the severity of the exposure, to ten types of maltreatment (emotional neglect, physical neglect, parental physical maltreatment, parental verbal abuse, non-verbal emotional abuse, peer emotional abuse, peer physical bullying, sexual abuse, witnessing violence toward siblings and witnessing interparental violence). Second, the MACE assesses the time period in which the maltreatment occurred. Each item asks for information about each year of childhood and adolescence, from 1 to 18. Research using the MACE has detected sensitive periods of maltreatment exposure associated with risk of psychopathology [21–23] as well as alterations in brain morphometry and function [24, 25]. This information could be useful for delineating periods of vulnerability to the most negative effects of maltreatment.

The MACE has shown excellent reliability and validity in U.S. samples [26]. The authors of the MACE have also made available the original set of 75 items, called the MACE-X, from which they extracted a 52 item US version of the MACE. From the larger set of items, researchers in multiple countries have been able to produce versions of the MACE appropriate for the

local culture. The U.S., Norwegian, and German versions [26–28] of the MACE have demonstrated excellent reliability and validity. To our knowledge, this scale has not been introduced into China and owing to cultural differences across countries, it is necessary to explore whether the MACE is a valid measure of retrospective reports of childhood maltreatment in this cultural context. Thus, this study aimed to (1) establish the MACE-CH, the Chinese version of the MACE scale, and (2) test its psychometric properties among a general sample of university students ages 18 to 26. Rasch modeling, test-retest reliability, convergent validity and predictive validity would be examined.

## Methods

### Procedure

Permission for the study was obtained from the Research Ethics Committee of the authors' university. The recruitment procedure began by handing out a brochure describing the project, making announcements and giving invitations to students. Participation in the study was entirely voluntary, and participants gave written consent to be part of the research. On the day of assessment, students were invited to participate voluntarily in their classes. Trained researchers administered the self-report questionnaires to students during class time. During data collection at baseline, we asked participants whether they could complete the MACE-X again after three months. Then 109 subjects completed the MACE-X once more to assess test-retest reliability. Participants were told that they should respond to the questionnaire items by themselves and that they were free to withdraw at any time during data collection. The confidentiality of the study was emphasized at the beginning of collection sessions.

### Participants

Participants were 812 undergraduate students (ages 18 to 26, M = 19.99, SD = 1.21) from five universities in the provinces of Zhejiang, Anhui and Guangdong in China. These students completed a translated version of the 75 MACE-X items as the basis for developing MACE-CH. Their demographic characteristics reflected those of students in these four universities. They came from diverse parts of the area: rural areas (32.14%), county seats (31.24%), small-medium cities (22.79%), and metropolitan areas (13.83%). Most reported financial sufficiency during childhood, defined as the amount of money available to the family: much less than enough (6.92%), less than enough (22.22%), enough (65.36%), more than enough (5.23%), and much more than enough (0.26%). A subsample of 513 students (30.66% males; ages 18 to 26, M = 19.84, SD = 1.21) also completed CTQ [29], QACE-R [16] and Depression Anxiety Stress Scales [30] at the beginning of the study. Of these, 109 completed the Chinese version of MACE-X again three weeks later to assess test-retest reliability.

### Initial items

The 75 item MACE-X assesses ten types of childhood maltreatment experienced before age 18. The types of maltreatment by parents or other adults were emotional neglect, physical neglect, physical maltreatment, verbal abuse, non-verbal emotional abuse, sexual abuse, witnessing interparental violence, and witnessing violence to siblings. The types of maltreatment by peers were peer emotional abuse and peer physical bullying. On each item, the respondent indicates "yes" or "no" separately for each year of exposure between the ages of 1 to 18. For the purposes of this study, the original English-language version of the MACE-X was translated into Chinese by three graduates who were fluent in both English and Chinese and knowledgeable about childhood adversity research. Further, we conducted a pilot study of 20 subjects to test if

participants had any difficulty in understanding or responding to the items in the translated version of the MACE-X. Items that were identified as problematic in the pilot study were discussed and modified. Finally, the updated version was back-translated into English and was compared with the original English version to confirm that the Chinese translated version of the MACE-X was consistent with the English version.

## Evaluation for item inclusion in a subscale

We aimed to develop ten subscales for the Chinese version of the MACE, corresponding to the ten types of maltreatment assessed by the MACE-X. Similarly, for statistical purposes we aimed for each subscale to include at least four items, and for these items to correspond to the MACE-X subscale items. A simple Rasch model was used to determine if final items measured the latent trait for each subscale. Rasch modeling was conducted in R version 4.0.3 with the eRm package [31] and ltm package [32].

We evaluated item fit using infit and outfit, which are the most widely used diagnostic Rasch mean-square fit statistics. Infit and outfit are calculated as the average squared residual for each person-item combination. Infit depicts unexpected responses to items with a difficulty close to the individual exposure levels, whereas Outfit is an outlier-sensitive fit that depicts responses to items with difficulty far from the individual. Acceptable ranges for mean square fit values are still up for debate. In the current analyses, outfit and infit values within the range of 0.50 to 1.50 were considered acceptable [33]. We plotted the test information function, which integrates the individual item information curves, in order to illustrate how well the information provided by each subscale identified participants in a category.

After items with acceptable Infit and Outfit were selected, we examined the measurement invariance using Andersen's Likelihood Ratio (LR) test [34]. Participants were placed in one of two groups based on a median-split on age among 812 subjects or on gender among 513 subjects (gender information was not collected for other 300 subjects). When the Anderson LR test is not significant, the item parameter estimates in Rasch modeling are invariant across two or more groups.

## Scoring algorithms and cutoffs

Based on the number of items positively endorsed by participants and person parameters in the Rasch model, subscales that included 5 or more items were scored for severity of exposure to the latent category. Typically these mean-centered logit scores range from -4 to +4. For the purposes of the current study they were further recalibrated to range from 0–10, enabling total exposure severity levels across the 10 subscales to fall in the range of 0 to 100. Because a Rasch model with only 4 items cannot export sufficient person parameters, subscales that contained 4 items were scored 0, 3, 5, 8 and 10 determined by a linear interpolation of the number of items that the participant positively endorsed.

The MACE and the QACE-R have nine overlapping categories (i.e., sexual abuse, emotional neglect, physical maltreatment, emotional abuse, physical neglect, witnessing interparental violence, peer emotional abuse and peer physical abuse). According to the procedures used to develop the English-language MACE, criterion (threshold) scores on the Chinese MACE to indicate exposure to these nine overlapping categories were created by comparing MACE severity scores to ACE (comparator) scores among 513 subjects. The R package "OptimalCut-points," which is based on ROC analyses, was used to select the optimal cutpoint for each subscale. Due to there being no corresponding subscale on the ACE, we operationally defined presence versus absence of witnessing violence to siblings as one selected item on the subscale.

## Assessment of reliability and validity

To evaluate the test-retest reliability, 109 participants from the original sample again completed the items in the Chinese MACE three weeks later. Test-retest reliability was assessed using Pearson correlations among Chinese MACE overall total scores, the 10 subscale total scores, and the Chinese MACE total score in each age group. Convergent validity was assessed by comparing the Chinese MACE scores to the QACE-R and CTQ scores using Pearson correlations.

## Assessment of utility as predictor of anxiety and depressive symptom scores

The utility of the Chinese MACE was then tested by examining its power in predicting scores on the Chinese version of the Depression Anxiety Stress Scales (DASS), a measure of self-rated stress, anxiety and depressive symptoms [30]. This measure has shown good reliability and validity in prior research [35]. For the purposes of this study, we used the anxiety and depression subscales as the outcome variables. Participants were instructed to assess their anxiety symptoms (e.g., "I was aware of dryness of my mouth") or depressive symptoms (e.g., "I found it difficult to work up the initiative to do things.") during the past week. Fourteen items (seven for each) were rated on a four-point scale ranging from 1 (did not apply to me at all) to 4 (applied to me very much of the time). The responses were averaged across items, with higher scores indicating higher anxiety or depressive symptoms. Consistent with a study by Teicher and Parigger (2015), we tested the Chinese MACE's Multiplicity scores' predictive power in relation to the QACE-R scores, and tested the MACE Severity scores' predictive power in relation to the total CTQ scores. First, ordinary least squares regression was performed to calculate the relationship between the MACE scores and symptom ratings. By this way, we can ascertain whether MACE had significantly stronger or weaker predictive power than the comparator scale (i.e., QACE-R or CTQ). Then the R relaimpo package was used to conduct multiple regression analysis with variance decomposition. This analysis provided a more precise determination of the percentage of variance in the DASS scores that was explained by scores on the MACE and scores on the comparator scales. Gender, age and financial sufficiency were included as covariates to control for the confounding influence of sociodemographic variables.

## Results

### Rasch analyses of ten subscales

**Parental physical maltreatment.** All six items of the considered items were included in this scale. These items had acceptable outfit and infit mean-square values (Table 1). None of the items had a fit index exceeding 1.5. However, one item had an outfit mean square fit less than 0.7. Fig 1 in S1 File shows the Item Characteristic Curves (ICC), Item Information Curves

**Table 1. Rasch analysis of parental physical maltreatment subscale.**

| MACE items | Item difficulty β (SE) | Outfit MSQ | Infit MSQ |
|---|---|---|---|
| 7. Intentionally pushed, pinched, slapped, kicked etc. | 0.64 (0.10) | 0.88 | 0.90 |
| 8. Hit you so hard it left marks for more than a few minutes | -0.66 (0.09) | 0.66 | 0.76 |
| 9. Hit or harmed you so severely that it needed medical attention | 2.05 (0.14) | 0.81 | 0.82 |
| 10. Spanked you on buttocks, arms or legs | -2.47 (0.14) | 0.91 | 0.84 |
| 11. Spanked you on unclothed buttocks | 0.92 (0.10) | 1.05 | 0.99 |
| 12. Spanked you with object such as belt, paddle, etc. | -0.47 (0.10) | 0.86 | 0.89 |

**Table 2. Rasch analysis of parental verbal abuse subscale.**

| MACE items | item difficulty β (SE) | Outfit MSQ | Infit MSQ |
|---|---|---|---|
| 1. Swore at you, called you names, insulted | 0.50 (0.09) | 0.91 | 0.93 |
| 2. Said hurtful things, made you feel humiliated | -0.84 (0.09) | 0.70 | 0.78 |
| 3. Yelled or screamed at you | -1.37 (0.10) | 0.74 | 0.85 |
| 4. Acted in a way that made you afraid that you might be physically hurt | 0.58 (0.09) | 1.02 | 0.97 |
| 5. Threatened to leave or abandon you | 1.13 (0.10) | 1.25 | 1.04 |

(IIC) and Test Information Function for the Rasch parental physical maltreatment subscale. Logit scores for the latent trait in the Rasch model ranged from -3.57 (no items selected) to 3.39 (all items selected). The Test Information Function indicated that the scale is more reliable in predicting medium to high exposure levels, with 37.03% of the overall information between logit scores of 0–2. Andersen's likelihood ratio test was not significant when splitting for age, $\chi^2(5) = 7.427$, $p = .191$, indicating acceptable fit to the Rasch model. Andersen's likelihood ratio test also showed no difference for males and females, $\chi^2(5) = 3.080$, $p = .688$. Based on ROC analyses, endorsement of at least two of the six selected items served as the operational definition of the presence of parental physical maltreatment.

**Parental verbal abuse.** This scale consisted of all five of the considered items. The items had acceptable infit and outfit, with mean square values ranging from 0.70–1.25 (Table 2). Fig 2 in S1 File shows the ICC, IIC and Test Information Function for the Rasch parental verbal abuse scale. Logit scores of the latent trait in the Rasch model ranged from -2.88 (no items selected) to 2.81 (all items selected). The Test Information Function indicated that the scale was best at discriminating medium to high exposure levels with 42.38% of the overall information between logit scores of 0–2. Further, Andersen's likelihood ratio test showed that the Rasch model did not significantly differ across the two age groups, $\chi^2(4) = 5.262$, $p = .261$. Andersen's likelihood ratio test was also not significant when splitting for gender, $\chi^2(4) = 8.972$, $p = .062$. Based on ROC analyses, a threshold was set at two of the five selected items to operationally define presence versus absence of parental verbal abuse.

**Non-verbal emotional abuse.** This scale included five of six considered items. One item with excessively high mean square outfit was eliminated. The remaining five items had acceptable infit and outfit mean square values, ranging from 0.83–1.01 (Table 3). Fig 3 in S1 File shows the ICC, IIC and Test Information Function for the Rasch non-verbal emotional abuse scale. Logit scores of the items representing the latent trait in the Rasch model ranged from -2.88 (no items selected) to 2.99 (all items selected). The Test Information Function indicated that the scale was best at discriminating for medium to high exposure levels with 37.69% of the overall information having logit scores between 0 and 2. The Andersen test was not significant when splitting for age, $\chi^2(4) = .773$, $p = .942$. When comparing males and females, the

**Table 3. Rasch analysis of parental non-verbal emotional subscale.**

| MACE items | Item difficulty β (SE) | Outfit MSQ | Infit MSQ |
|---|---|---|---|
| 6. Locked you in closet, basement, garage, etc. | 1.74 (0.14) | 0.99 | 0.9 |
| 55. Parent very difficult to please | 0.51 (0.10) | 1.01 | 0.93 |
| 65. Had to shoulder adult responsibilities | -0.89 (0.09) | 0.87 | 0.9 |
| 66. Felt family financial pressure | -1.17 (0.09) | 0.83 | 0.88 |
| 67. Kept important secrets/facts from you | -0.18 (0.09) | 0.95 | 0.96 |

**Table 4. Rasch analysis of peer emotional abuse subscale.**

| MACE items | Item difficulty β (SE) | Outfit MSQ | Infit MSQ |
|---|---|---|---|
| 39. Swore, called you names/insults more than few times per year | -0.1 (0.09) | 0.91 | 0.93 |
| 40. Said hurtful things made you feel humiliated more than few times per year | -0.62 (0.09) | 0.72 | 0.81 |
| 41. Said things behind you back, spread rumors | -1.11 (0.09) | 0.88 | 0.98 |
| 42. Excluded you from activities / groups | 0.26 (0.09) | 0.92 | 0.95 |
| 43. Acted in way that made you afraid you might be hurt | 1.57 (0.11) | 1.15 | 1.04 |

Andersen test was also not significant, $\chi^2(4) = 4.561$, $p = .335$. A threshold was set at one selected item to operationally define presence versus absence of non-verbal emotional abuse.

**Peer emotional abuse.** All five considered items showing acceptable mean square infit and outfit statistics were included in this scale (Table 4). Fig 4 in S1 File shows the ICC, IIC and Test Information Function for the Rasch peer emotional abuse scale. Logit scores of the latent trait in the Rasch model ranged from -2.77 (no items selected) to 2.86 (all items selected). The Test Information Function indicated that the scale was best at discriminating for medium to high exposure levels with 42.08% of the overall information between logit scores of 0–2. The Andersen test was not significant when splitting for age, $\chi^2(4) = 2.877$, $p = .579$. When comparing males and females, the Andersen test was significant, $\chi^2(4) = 49.548$, $p < .001$. The Wald test indicated differential response patterns for males and females for item 41 (Z = 3.264, $p = .001$), item 42 (Z = 3.842, $p < .001$) and item 43 (Z = -5.742, $p < .001$). A threshold was set at two selected items to operationally define presence versus absence of peer emotional abuse.

**Peer physical bullying.** All five considered items were included in this scale. The infit and outfit mean squares ranged from 0.66–1.14 (Table 5). Fig 5 in S1 File shows the ICC, IIC and Test Information Function for the Rasch peer physical bullying scale. Logit scores of the latent trait in the Rasch model ranged from -2.97 (no items selected) to 3.19 (all items selected). The overall Test Information Function indicated that this test provided the most information in the high exposure level of the trait with 37.9% of the overall information between logit scores of 2–4. The Andersen test was not significant when splitting for age, $\chi^2(4) = 8.806$, $p = .066$. When comparing males and females, the Andersen test was significant, $\chi^2(4) = 11.569$, $p = .021$. The Wald test indicated differential response patterns for males and females for item 45 (Z = 2.854, $p = .004$). A threshold was set at one selected item to operationally define presence versus absence of peer physical bullying.

**Sexual abuse.** This scale consisted of eight of the twelve considered items involving adult familial, adult extrafamilial as well as peer sexual abuse. The infit and outfit mean squares of the remaining items ranged from 0.53 to 1.31 (Table 6). Fig 6 in S1 File showed the ICC, IIC and Test Information Function for the Rasch sexual abuse scale. Logit scores of the latent trait

**Table 5. Rasch analysis of peer physical bullying subscale.**

| MACE items | Item difficulty β (SE) | Outfit MSQ | Infit MSQ |
|---|---|---|---|
| 44. Threatened you in order to take money or possessions | 0.35 (0.14) | 0.96 | 0.96 |
| 45. Forced you to do things you did not want to | -1.38 (0.12) | 1.14 | 1.13 |
| 46. Intentionally pushed, shoved, punched, kicked you etc. | -0.88 (0.12) | 0.82 | 0.86 |
| 47. Hit you so hard it left marks for more than a few minutes | -0.20 (0.13) | 0.76 | 0.8 |
| 48. Hit or harmed you so severely as to need medical attention | 2.11 (0.24) | 0.66 | 0.73 |

**Table 6. Rasch analysis of sexual abuse subscale.**

| MACE items | Item difficulty β (SE) | Outfit MSQ | Infit MSQ |
|---|---|---|---|
| 13. Parents inappropriate sexual comments to you | 0.63 (0.26) | 1.11 | 1.07 |
| 14. Parents touched or fondled you in sexual way | 1.00 (0.30) | 1.31 | 1.06 |
| 26. Other adults' inappropriate sexual comments to you | -1.41 (0.17) | 1.09 | 1.02 |
| 27. Other adults touched or fondled you in sexual way | -1.03 (0.18) | 0.87 | 0.89 |
| 28. Other adults had sexual intercourse with you | 0.09 (0.23) | 0.53 | 0.66 |
| 29. Other adults attempted to have any type of sexual intercourse with you | 0.55 (0.26) | 0.66 | 0.77 |
| 49. Peer(s) forced you to engage in sexual activity against your will | -0.45 (0.20) | 1.31 | 1.24 |
| 50. Peer(s) forced you to do things sexually you did not want to do | 0.63 (0.26) | 0.77 | 0.85 |

in the Rasch model ranged from -3.21 (no items selected) to 3.14 (all items selected). The overall Test Information Function indicated that this test provided the most information in the high exposure level of the trait with 35.66% of the overall information between logit scores of 2–4. The Andersen test was not significant when splitting for age, $\chi^2(7) = 3.176$, $p = .868$. When comparing males and females, the Andersen test was also not significant, $\chi^2(7) = 11.332$, $p = .125$. A threshold was set at one selected item to operationally define presence versus absence of sexual abuse.

**Witnessing violence to siblings.** Four of the eight considered items involving witnessing physical and/or sexual abuse to siblings were included in this scale (Table 7). These items provided the best overall fit. Fig 7 in S1 File shows the ICC, IIC and Test Information Function for the Rasch scale for witnessing violence to siblings. The overall Test Information Function indicated that this test provided the most information in the high exposure level of the trait with 37.49% of the overall information between logit scores of 2–4. The Andersen test was not significant when splitting for age, $\chi^2(3) = 4.281$, $p = .233$. When comparing males and females, the Andersen test was significant, $\chi^2(3) = 9.366$, $p = .025$. The Wald test indicated differential response patterns for males and females for item 19 (Z = 2.695, $p = .007$) and item 25 (Z = -2.368, $p = .018$).

**Witnessing interparental violence.** This scale consisted of six of eight considered items. Two items were removed because of high outfit mean square fit. The remaining items showed acceptable infit and outfit mean square values that ranged from 0.65 to 1.35 (Table 8). Fig 8 in S1 File shows the ICC, IIC and Test Information Function for the Rasch witnessing interparental violence scale. Logit scores of the latent trait in the Rasch model ranged from -3.64 (no items selected) to 3.86 (all items selected). The overall Test Information Function indicated that this test provided the most information in the high exposure level of the trait with 38.88%

**Table 7. Rasch analysis of witnessing violence to siblings subscale.**

| MACE items | Item difficulty β (SE) | Outfit MSQ | Infit MSQ |
|---|---|---|---|
| 18. Intentionally pushed, grabbed, shoved, slapped, pinched, punched, or kicked your sibling | -0.74 (0.13) | 0.74 | 0.77 |
| 19. Hit your sibling so hard that it left marks for more than a few minutes | -1.79 (0.15) | 0.93 | 0.89 |
| 20. Parents hit or intentionally harmed sibling so that they needed medical attention | 0.91 (0.16) | 0.69 | 0.74 |
| 25. Threatened to harm your sibling | 1.62 (0.20) | 0.78 | 0.75 |

**Table 8. Rasch analysis of witnessing interparental violence subscale.**

| MACE items | Item difficulty β (SE) | Outfit MSQ | Infit MSQ |
|---|---|---|---|
| 33. Saw adults living in household push, slap or throw something at mother (stepmother, grandmother) | -2.41 (0.18) | 1.35 | 0.88 |
| 34. Saw adults hit mother (or surrogates) so hard that it left marks for more than a few minutes | -1.13 (0.17) | 0.84 | 0.89 |
| 35. Saw adults hit or harm mother (or surrogates) to the extent that it needed medical attention | 0.75 (0.21) | 0.65 | 0.89 |
| 36. Saw adults living in household push, slap or throw something at father (stepfather, grandfather) | -0.60 (0.17) | 0.91 | 0.95 |
| 37. Saw adults hit father (or surrogates) so hard that it left marks for more than a few minutes | 0.53 (0.20) | 0.66 | 0.74 |
| 38. Saw adults hit or harm father (or surrogates) to the extent that it needed medical attention | 2.86 (0.39) | 1.16 | 0.70 |

of the overall information between logit scores of 2–4. The Andersen test was not significant when splitting for age, $\chi^2(5) = 5.625$, $p = .344$. When comparing males and females, item 38 didn't meet the requirement to be included in the Andersen test. For the remaining items, the Andersen test was also not significant when splitting for gender, $\chi^2(4) = 1.822$, $p = .769$. A threshold was set at one selected item to operationally define presence versus absence of witnessing interparental violence.

**Emotional neglect.** This scale included eight of the nine considered items. Only one item with excessively high mean square outfit was removed. The remaining eight items had acceptable infit and outfit mean square values ranging from 0.75–1.16 (Table 9). Four of the items were reverse scored (items 57, 58, 74, 75 on the MACE-X). Fig 9 in S1 File shows the ICC, IIC and Test Information Function for the Rasch emotional neglect scale. Logit scores of the latent trait in the Rasch model ranged from -2.88 (no items selected) to 2.88 (all items selected). The Test Information Function indicated that the scale was best at discriminating for medium to high exposure levels with 38.85% of the overall information between logit scores of 2–4 and with 36.98% of the overall information between logit scores of 0–2. The Andersen test was not significant when splitting for age, $\chi^2(7) = 10.521$, $p = 0.161$. When comparing males and females, the Andersen test was significant, $\chi^2(7) = 17.305$, $p = .016$. The Wald test indicated differential response patterns for males and females for item 57 (Z = -2.579, $p = .010$) and item 53 (Z = 2.675, $p = .007$). A threshold was set at one selected item to operationally define presence versus absence of emotional neglect.

**Table 9. Rasch analysis of emotional neglect subscale.**

| MACE items | Item difficulty β (SE) | Outfit MSQ | Infit MSQ |
|---|---|---|---|
| 51. Mother unavailable poor reasons | 0.01 (0.10) | 1.16 | 1.08 |
| 52. Father unavailable poor reasons | -0.02 (0.10) | 0.98 | 0.98 |
| 53. Mother good poor reasons | 0.18 (0.11) | 0.77 | 0.82 |
| 54. Father unavailable poor reasons | 0.20 (0.11) | 0.75 | 0.83 |
| 57. Family member made you feel loved (reversed) | 0.22 (0.11) | 0.98 | 1.01 |
| 58. Family member helped you feel special/important (reversed) | -0.48 (0.09) | 1.14 | 1.14 |
| 74. People in your family felt close to each other | 0.04 (0.10) | 1.05 | 1.04 |
| 75. Your family was a source of strength and support | -0.16 (0.10) | 1.05 | 1.06 |

**Table 10. Rasch analysis of physical neglect subscale.**

| MACE items | Item difficulty (SE) | Outfit MSQ | Infit MSQ |
|---|---|---|---|
| 60. One or more would be there to take you to doctor or ER if needed (reverse) | -0.20 (0.11) | 0.74 | 0.81 |
| 61. One or more would be there to help you with your homework, or to get ready for school. | -0.81 (0.10) | 0.87 | 0.88 |
| 62. You did not have enough to eat | 0.74 (0.13) | 0.95 | 0.94 |
| 63. You had to wear dirty clothes | 1.11 (0.15) | 0.85 | 0.86 |
| 64. You were left unsupervised at an age or in situations when you should have been supervised | -0.79 (0.10) | 1.35 | 1.34 |
| 73. People in family looked out for each other (reverse) | -0.04 (0.11) | 0.90 | 0.91 |

**Physical neglect.** Six of seven considered items were included in this scale. These items showed acceptable infit and outfit mean square values ranging from 0.74–1.35 and provided the best overall fit (Table 10). Three of the items were reverse scored (items 60, 61, 73 on the MACE-X). Fig 10 in S1 File shows the ICC, IIC and Test Information Function for the Rasch physical neglect scale. Logit scores of the latent trait in the Rasch model ranged from -2.82 (no items selected) to 2.84 (all items selected). The Test Information Function indicated that the scale was best at discriminating for medium to high exposure levels with 40.04% of the overall information between logit scores of 2–4. The Andersen test was not significant when splitting for age, $\chi^2(5) = 4.951$, $p = 0.422$. When comparing males and females, the Andersen test was not significant, $\chi^2(5) = 7.231$, $p = .204$. A threshold was set at one selected item to operationally define presence versus absence of physical neglect.

## Test–retest reliability

A subsample of 109 volunteers participated in the retest three weeks later. Using the Chinese version of the MACE with 58 items (S2 File) on the basis of the analyses above, test-retest reliability was excellent (defined as $r > 0.8$) for total scores ($r = 0.887$, $p < 10^{-16}$), multiplicity scores ($r = 0.835$, $p < 10^{-16}$), duration scores ($r = 0.812$, $p < 10^{-16}$), overall degree of exposure to parental physical maltreatment ($r = 0.885$, $p < 10^{-16}$), overall degree of exposure to witnessing interparental violence ($r = 0.865$, $p < 10^{-16}$), and overall degree of exposure to emotional neglect ($r = 0.821$, $p < 10^{-16}$). Test-retest reliability was very good (defined as $0.7 < r < 0.8$) for the overall scores of exposure to the seven other types of maltreatment (Table 11).

**Table 11. Test–retest reliability by types.**

| Type of maltreatment | Test-Retest r | Confidence Interval |
|---|---|---|
| Parental Physical Maltreatment. | 0.885 | 0.836–0.920 |
| Parental Verbal Abuse | 0.798 | 0.717–0.857 |
| Non-Verbal Emotional Abuse | 0.790 | 0.707–0.851 |
| Sexual Abuse | 0.750 | 0.655–0.822 |
| Peer Emotional Abuse | 0.731 | 0.630–0.808 |
| Peer Physical Bullying | 0.707 | 0.599–0.790 |
| Witnessing Interparental Violence | 0.865 | 0.809–0.906 |
| Witnessing Violence to Siblings | 0.718 | 0.613–0.798 |
| Emotional neglect | 0.821 | 0.748–0.874 |
| Physical Neglect | 0.775 | 0.687–0.841 |

**Table 12. Test–retest reliability by age.**

| Recollected Ages | Test-Retest r | Confidence Interval |
|---|---|---|
| 1 | 0.815 | 0.740–0.870 |
| 2 | 0.822 | 0.750–0.875 |
| 3 | 0.800 | 0.720–0.859 |
| 4 | 0.846 | 0.782–0.892 |
| 5 | 0.862 | 0.803–0.904 |
| 6 | 0.777 | 0.689–0.842 |
| 7 | 0.755 | 0.661–0.826 |
| 8 | 0.811 | 0.735–0.867 |
| 9 | 0.814 | 0.739–0.869 |
| 10 | 0.870 | 0.815–0.909 |
| 11 | 0.835 | 0.767–0.884 |
| 12 | 0.825 | 0.754–0.877 |
| 13 | 0.777 | 0.690–0.842 |
| 14 | 0.798 | 0.717–0.857 |
| 15 | 0.744 | 0.646–0.817 |
| 16 | 0.819 | 0.745–0.872 |
| 17 | 0.825 | 0.753–0.877 |
| 18 | 0.774 | 0.686–0.840 |

Similarly, test-retest reliability for MACE total scores at each age from years 1 to 18 showed very good to excellent reliability (Table 12).

## Convergent validity

A subsample of 513 participants completed the Chinese MACE and two other maltreatment scales (QACE-R and CTQ) at the beginning of the study. Tests of convergent validity were conducted by assessing the correlations among the Chinese MACE scores (Total score and Multiplicity score) and scores on the QACE-R and CTQ. The QACE-R score was highly correlated with both the MACE Total score ($r = 0.738$, $p < 10^{-16}$) and the MACE Multiplicity score ($r = 0.706$, $p < 10^{-16}$). CTQ scores showed moderate to high correlations with the MACE Total score ($r = 0.584$, $p < 10^{-16}$) and the MACE Multiplicity score ($r = 0.569$, $p < 10^{-16}$). As expected, the MACE Total score was highly correlated with the MACE Multiplicity score ($r = 0.934$, $p < 10^{-16}$). The correlations between each MACE subscale score and the corresponding CTQ and QACE-R subscale scores were all significant ($p < 10^{-16}$) and in the expected direction (Table 13).

## Comparison of MACE, QACE-R and CTQ scores as predictors of anxiety and depression (DASS)

We tested the predictive validity of the Chinese MACE in the same subsample of 513 participants. The MACE Severity score and the CTQ score were compared in terms of the strength of their associations with the DASS anxiety and depression scores. The MACE Multiplicity score and the QACE-R score were similarly compared using Ordinary Least Squares regression (Tables 14 and 15). The Williams test showed that anxiety was significantly ($p = .05$) more strongly correlated with the MACE Total score ($r = 0.27$) than with the CTQ score ($r = 0.12$). There was a significant difference in the strength of the correlation between depression and the MACE Total score ($r = 0.27$) and the correlation between depression and the CTQ score

**Table 13. Pearson correlation coefficients between matching subscales of MACE and CTQ / ACE.**

| MACE subscales | CTQ subscale | ACE subscale |
|---|---|---|
| Parental physical abuse | Physical abuse (r = 0.561, p < $10^{-16}$) | Physical abuse (r = 0.588, p < $10^{-16}$) |
| Parental verbal abuse | Emotional abuse (r = 0.542, p < $10^{-16}$) | Emotional abuse (r = 0.551, p < $10^{-16}$) |
| Emotional neglect | Emotional neglect (r = 0.512, p < $10^{-16}$) | Emotional neglect (r = 0.382, p < $10^{-16}$) |
| Physical neglect | Physical neglect (r = 0.436, p < $10^{-16}$) | Physical neglect (r = 0.409, p < $10^{-16}$) |
| Parental non-verbal emotional abuse | Emotional abuse (r = 0.494, p < $10^{-16}$) | Emotional abuse (r = 0.357, p < $10^{-16}$) |
| Sexual abuse | Sexual abuse (r = 0.439, p < $10^{-16}$) | Sexual abuse (r = 0.568, p < $10^{-16}$) |

($r$ = 0.16). The MACE Multiplicity score and the QACE-R did not differ in terms of the strength of their association with either anxiety ($r$ = 0.26 and $r$ = 0.24, respectively) or depression ($r$ = 0.23 and $r$ = 0.26, respectively). Two multiple regression analyses with gender, age and financial sufficiency as covariates revealed significant effects of MACE Severity and Multiplicity scores on depressive and anxiety symptoms. Further, variance decomposition analyses found that MACE Severity scores explained more variance in symptom ratings of both anxiety and depression than the CTQ scores did (Table 14), and MACE Multiplicity scores explained more variance in symptom ratings of anxiety than the QACE-R scores did (Table 15).

## Discussion

The MACE is a commonly used scale used to assess exposure to maltreatment. In this study we created a Chinese version of the MACE (MACE-CH). Scale development was based on the Chinese translation of the MACE-X [26], a set of items that can be adapted to create culture-specific versions of the scale. The Chinese version includes 58 of the 75 MACE-X items, and retained the ten-subscale structure of the MACE. The scale showed good internal consistency,

**Table 14. Comparative difference between MACE SUM and CTQ scores.**

| | Depressive symptoms (n = 404) | | Anxiety symptoms (n = 404) | |
|---|---|---|---|---|
| | Ordinary Least Squares | | | |
| | r | p-value | r | p-value |
| MACE SUM | 0.27 | < $10^{-7}$ | 0.27 | < $10^{-7}$ |
| CTQ Scores | 0.16 | < $10^{-2}$ | 0.12 | 0.02 |
| Williams Test (z) | 1.64 | 0.05 | 2.21 | 0.01 |
| | Multiple Regression | | | |
| | Beta | p-value | Beta | p-value |
| MACE SUM | 0.261 | < $10^{-4}$ | 0.303 | < $10^{-6}$ |
| CTQ Scores | 0.007 | 0.901 | -0.066 | 0.271 |
| Gender | -0.000 | 0.998 | 0.032 | 0.512 |
| Age | 0.076 | 0.125 | 0.107 | 0.030 |
| Financial Sufficiency | 0.073 | 0.144 | 0.060 | 0.225 |
| | Variance Decomposition | | | |
| MACE SUM | 5.86% | | 6.69% | |
| CTQ Scores | 1.29% | | 0.78% | |
| Gender | 0.00% | | 0.09% | |
| Age | 0.69% | | 1.30% | |
| Financial Sufficiency | 0.33% | | 0.19% | |

**Table 15. Comparative difference between MACE MULTI and ACE scores.**

| | Depressive symptoms (n = 492) | | Anxiety symptoms (n = 492) | |
|---|---|---|---|---|
| | Ordinary Least Squares | | | |
| | r | p-value | r | p-value |
| MACE MULTI | 0.23 | $<10^{-5}$ | 0.26 | $<10^{-6}$ |
| ACE Scores | 0.26 | $<10^{-6}$ | 0.24 | $<10^{-5}$ |
| Williams Test (z) | 0.50 | 0.31 | 0.33 | 0.37 |
| | Multiple Regression | | | |
| | Beta | p-value | Beta | p-value |
| MACE MULTI | 0.100 | 0.134 | 0.170 | 0.011 |
| ACE Scores | 0.192 | 0.004 | 0.119 | 0.075 |
| Gender | 0.009 | 0.845 | 0.035 | 0.450 |
| Age | 0.091 | 0.047 | 0.101 | 0.020 |
| Financial Sufficiency | 0.081 | 0.101 | 0.068 | 0.170 |
| | Variance Decomposition | | | |
| MACE MULTI | 2.93% | | 3.95% | |
| ACE Scores | 4.17% | | 3.08% | |
| Gender | 0.03% | | 0.16% | |
| Age | 0.97% | | 1.41% | |
| Financial Sufficiency | 0.36% | | 0.23% | |

three-week test-retest reliability, convergent validity with other maltreatment scales, and validity in predicting anxiety and depression. Based on Rasch modeling, each of the ten subscales consisted of 4–9 items. Subscales that had five or more items were scored for severity of exposure using latent logit scores in a Rasch model. However, in the current Chinese MACE, there were only four items in the subscale assessing witnessing violence to siblings. These items could not cover the latent trait by using logit scores. Instead, they were scored 0, 3, 5, 8 and 10 according to a linear interpolation of how many items were positively endorsed. The other nine subscales with 5 or more items were scored using logit scores.

The test-retest reliability for the Chinese MACE subscale scores was similar to that found in the original U.S. version of the MACE [26]. However, the time interval from test to retest was much shorter in our study—three weeks compared to 66 days on average in Teicher and Parigger's (2015) study of the original MACE. The shorter time lag may account for the numerically higher test-retest reliability values for the physical and emotional neglect subscales in the Chinese MACE compared to the U.S. MACE. The test-retest reliability of the MACE severity scores for reports of maltreatment from age 1 to age 4 was also higher in the Chinese MACE than in the U.S. MACE. On the other hand, test-retest reliability in the current study was slightly lower than that in the Norwegian MACE [27] although the time interval from test to retest was very similar. The difference in test-retest reliability may have to do with the nature of the two samples. The sample in the Norwegian scale development study included both psychiatric outpatients and employees, and the average MACE score was high. By contrast, our study was conducted in a sample of healthy undergraduates.

Convergent validity was tested by comparing the Chinese MACE to two other measures of maltreatment, namely the CTQ and QACE-R. The three measures were moderately inter-correlated, suggesting that all three can be said to have convergent validity in measuring retrospective accounts of maltreatment. However, research on the original versions of these three measures showed that compared to the CTQ and ACE instruments, the MACE explained twice as much variance in psychiatric symptoms [26]. The Norwegian MACE also showed

better predictive validity than the CTQ on SCL-90 in patients with moderately severe mental health problems. Our results were consistent with this earlier evidence. The Chinese MACE scores showed a significantly stronger correlation with depressive and anxiety symptoms than the CTQ and QACE-R scores did. Moreover, results of variance decomposition indicated that MACE Total scores accounted for on average 6.28% of the variance in symptom ratings, whereas the CTQ accounted for 1.04%. The MACE Multiplicity scores on average accounted for 1.28 times more of the variance in anxiety symptom ratings compared to the QACE-R score.

There is growing evidence that the most important determinants of psychological disorders is exposure to specific types of childhood maltreatment during specific windows of vulnerability [1, 36]. Using the U.S. version of MACE, Khan et al. (2015) reported that the most important predictor of suicidal ideation was parental verbal abuse at age 5 in males and sexual abuse at age 18 in females. A neuroimaging study in the U.S. also identified windows of vulnerability using the MACE to assess childhood maltreatment. The results showed that exposure to maltreatment during early childhood was significantly associated with blunted amygdala response, whereas early teen exposure was significantly associated with augmented amygdala response [25]. The availability of the Chinese MACE will allow similar research to be conducted in the Chinese cultural context.

This work presents four limitations that have to be considered. First, there is an imbalance in number of items per perpetrator within sexual abuse subscale. Only two items of parental sexual abuse were included, while for other adults 4 items were included. Other items of parental sexual abuse appear to be rarely reported events, contributing to very low Outfit MSQ. Thus, they are not included in final subscale of sexual abuse. It would be valuable in future to test whether there are different results when these items of parental sexual abuse are assessed in highly exposed individuals (e.g. individuals with mental illness). Second, the Chinese MACE subscale for witnessing violence to siblings had too few items to be represented by mean-centered logit scores. In future research, a subscale with a sufficient number of items could be further analyzed in a Rasch model. Third, the duration for testing the test-retest reliability is not ideal. Future work using a longer duration would be better. Last, in addition to the Depression and Anxiety Stress Scales, more scales rating psychopathology should be used to test the predictive power of MACE-CH.

## Conclusion

This research created the first Chinese version of the MACE as a self-report measure of childhood maltreatment. The Chinese version is not simply a translation of the original measure, but is instead a version of the MACE that is appropriate for research in the Chinese cultural context. Chinese MACE retained the ten-subscale structure of the MACE and showed good internal consistency, three-week test-retest reliability, convergent validity with other maltreatment scales, and validity in predicting anxiety and depression. The development of this measure is a key advancement in Chinese researchers' ability to identify the type and timing of childhood maltreatment. The MACE-CH will be provided as open access to help facilitate its free use.

## Supporting information

**S1 File. Figures from Rasch modeling for MACE-CH subscales.**
(DOCX)

**S2 File. The Chinese MACE instrument.**
(PDF)

**S3 File. Chinese MACE-X data file.**
(CSV)

**S4 File. Test-retest data file.**
(CSV)

## Acknowledgments

We are very grateful to Dr. Martin H. Teicher (McLean Hospital, USA) for his permission and help.

## Author Contributions

**Conceptualization:** Yuanyuan Chen, Jianjun Zhu.

**Formal analysis:** Yuanyuan Chen.

**Funding acquisition:** Jianjun Zhu.

**Investigation:** Zhen Wang, Xiaoyu Zheng, Zhiyin Wu.

**Methodology:** Yuanyuan Chen, Zhen Wang, Jianjun Zhu.

**Supervision:** Jianjun Zhu.

**Visualization:** Yuanyuan Chen, Zhen Wang.

**Writing – original draft:** Yuanyuan Chen, Zhen Wang, Xiaoyu Zheng, Jianjun Zhu.

**Writing – review & editing:** Xiaoyu Zheng, Zhiyin Wu.

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
