## [Decision Letter · Decision Letter 0]

19 May 2022

PONE-D-22-05087The Chinese version of the Maltreatment and Abuse Chronology of Exposure (MACE) scale: Psychometric properties in a sample of young adultsPLOS ONE

Dear Dr. Zhu,

Thank you for submitting your manuscript to PLOS ONE. After careful consideration, we feel that it has merit but does not fully meet PLOS ONE’s publication criteria as it currently stands. Therefore, we invite you to submit a revised version of the manuscript that addresses the points raised during the review process. As you con see below, all reviewers expressed their enthusiasm for your manuscript. The reviewers recommended minor revisions, primarily to increase transparency and clarity of the paper.

We look forward to receiving your revised manuscript.

Kind regards,

Torsten Klengel, MD PhD

Academic Editor

PLOS ONE

Journal Requirements:

Reviewers' comments:

Reviewer's Responses to Questions

**Comments to the Author**

1. Is the manuscript technically sound, and do the data support the conclusions?

Reviewer #1: Yes

Reviewer #2: Yes

Reviewer #3: Yes

2. Has the statistical analysis been performed appropriately and rigorously? 

Reviewer #1: Yes

Reviewer #2: Yes

Reviewer #3: Yes

3. Have the authors made all data underlying the findings in their manuscript fully available?

Reviewer #1: Yes

Reviewer #2: Yes

Reviewer #3: Yes

4. Is the manuscript presented in an intelligible fashion and written in standard English?

Reviewer #1: Yes

Reviewer #2: Yes

Reviewer #3: Yes

5. Review Comments to the Author

Reviewer #1: The authors present a study where they have developed and psychometrically tested a Chinese version of the MACE trauma questionnaire. They first used Rasch modelling to develop a shorter version of the original US/German 75 item MACE-X, using data from a student sample, leading to a 58 item Chinese MACE. They proceed by investigating test-retest reliability of the MACE-CH-58, convergence of the MACE-CH-58 with two other trauma instruments, the CTQ and ACE-Q-R, and discriminant validity of their MACE-version compared to CTQ and ACE-Q-R in terms of variance explained in symptoms of anxiety and depression. The authors reported good psychometric properties of the MACE-CH-58, and superior discriminant validity as compared to the other trauma instruments.

The study is well conducted, carried out within an adequate methodological approach, using suitable statistical analysis, and with clear, well-structured introduction, methods, results presentation and discussion where results are adequately placed in context. The study represents a solid development and testing of the Chinese MACE and a valuable contribution to this important field of research.

Additional comments:

ABSTRACT

The following sentence is somewhat confusing: “Rasch modeling was used for scale development in a sample of 812 undergraduate students. Item reduction analysis of the original 75 items used to develop the 52-item MACE produced a 58-item Chinese version.” What is the “52-item MACE” that is referred to here? Either add information on what this version is (e.g. it is the standard US version), or remove the reference to this unidentified version. I would suggest the latter.

The authors write: “The new scale showed the same factor structure as the original MACE”. It is not clear to me what they mean here, since they did not perform any factor analysis to extract subdimensions on the MACE. Instead, the authors relied on the 10 existing dimensions of MACE and used Rasch analysis on each of the dimensions to reduce number of items. Indeed, in Methods they write “We aimed to develop ten subscales for the Chinese version of the MACE, corresponding to the ten types of maltreatment assessed by the MACE-X.”. Hence, it appears that the reference to factor structure of the Chinese MACE in the abstract, should be removed or reformulated.

INTRODUCTION

At the end of the introduction, the authors should specify which psychometric analysis they will perform: test-retest, convergent validity, discriminant validity etc.

METHODS

Methods section may start out with a paragraph that describes the study design.

Part 2.1 on participants: At the end of the paragraph, the authors write that “109 <participants> completed all measures again three weeks later to assess test-retest reliability”. I had thought that it was only MACE-CH that was administered again after three weeks, with the other instruments administered only at baseline?

RESULTS

At the end of the Results section, under the heading “Recollected time of exposure”, the authors present some data on age of exposure. I could not find any notice or description in Methods that such an analysis should be performed or how it can be understood. In results, some aspects of the methods are noted, but this should be moved to Methods. Moreover, it is not fully clear to me (it does not come clearly across) what the authors actually did, and which results they found in terms of ages of exposure – which they present in table 16. Did they find that, for each subdimension, certain ages occurred more often than others?</participants>

Reviewer #2: The study is about the psychometric validation of the Chinese MACE (Maltreatment and Abuse Chronology of Exposure), and the scale demonstrates high quality psychometric properties.

The findings are based on a larger sample size of students, and the manuscript is a value contribution to the literature.

General comment:

Please add a comment if you plan making the MACE-CH fully available (open access, upon request)

I do not have any major concerns, and thus; I would recommend a minor revision following more specific comments:

Abstract:

- The MACE-risk associations are not limited to psychopathology (“identify periods of development when childhood maltreatment maximally increases vulnerability to psychopathology“). Health outcomes?

- MACE can also be used in clinical settings and is not only limited to research.

Introduction:

- when you refer to the population-attributable risk. Do you mean the population-attributional risk fraction?

Methods/Results:

- Methods: please specify all important parameters that were set e.g. for random effects in linear mixed effect models.

- Please report how you have made sure, that participants reported events correctly (e.g., an event that happened at the age of “5”/”0” needs to be checked for the 6th /first year of life)….

- Please specify, which items were considered for potential inclusion of each subscales.

- How was the subsample selected that did the retest of the MACE-CH?

- Please explain why Rasch models cannot export sufficient person parameters? And why subscale with 4 items were not scored as 0 = 0, 1= 2.5, 2=5, 3 = 7.5 and 4 = 10?

- The age range of the present sample is restricted 18-26years. Would also another slitting would be interesting to detect differential responding for women and men?

Results:

- The authors seem to consider more strict ranges of infits and outfits for the items as compared to description in the methods (1.3 and 0.7 instead of 1.5 and 0.5). Please be consistent.

- I would recommend not to mention the scaled scores when reporting the number of positively endorsed items for cut off severity (Parental verbal abuse).

- Non-verbal emotional abuse: when “splitting”… instead of “slipping”.

- Sexual abuse: The df for the Andersen test was 6. Please report which item was dropped for this analysis. And please check the df for other reported tests (e.g., witnessed violence to siblings).

- Please also report test-retest reliability of the MACE duration (years with a multiplicity score ≥ 1 (ranging from 0 to 18).

Discussion:

- There is an imbalance in number of items per perpetrator. Only two items of parental sexual abuse were included, while for other adults 4 items were included. This means, that two potentially important items were not scored if endorsed (1) parents attempted to have any type of sexual intercourse with you (2) parents had sexual intercourse with you. From the item difficulties it appears to be rarely reported events. Please revise the subscale or discuss this issue. Would you expect different results when assessing these items in highly exposed individuals (e.g. individuals with mental illness)?

Table 12: Please specify the parameter of the MACE that you have used for this correlation.

Reviewer #3: Dear Authors,

Congratulations on this excellent manuscript and the successful development of the first Chinese version of the MACE. I truly appreciated the amount of due diligence taken at each step of the analyses. The manuscript has very closely followed the methodology proposed in the original publication by Teicher and Parigger. The writing has been excellent. The only thing worth correcting would be in the second last paragraph, line 3 of the discussion section, where the Khan et al paper should be referenced to 2015 and not 2017.

There were only two minor comments, for future considerations. The duration for testing the test-retest reliability which you have acknowledged as a limitation, was not ideal. Additionally, I wish you had used a scale or two in addition to the Depression and Anxiety Stress Scales.

Overall, this is an excellent manuscript and I believe could be used as a template by researchers interested in translating the MACE into local languages.

Warm regards.

6. PLOS authors have the option to publish the peer review history of their article (what does this mean?). If published, this will include your full peer review and any attached files.

Reviewer #1: **Yes: **Roar Fosse

Reviewer #2: **Yes: **Inga Schalinski

Reviewer #3: **Yes: **Alaptagin Khan

---

## [Author Response · Author response to Decision Letter 0]

30 May 2022

May 27, 2022

Torsten Klengel, MD PhD

Academic Editor

PLOS ONE

Manuscript Number: PONE-D-22-05087

Dear Dr. Klengel:

Thank you for the constructive feedback from you and the reviewers of our manuscript, “The Chinese version of the Maltreatment and Abuse Chronology of Exposure (MACE) scale: Psychometric properties in a sample of young adults”, Manuscript Number: PONE-D-22-05087. We have revised the manuscript and incorporated all suggestions and comments from you and reviewers. Below are our responses in detail.

Reviewer #1

1. The following sentence is somewhat confusing: “Rasch modeling was used for scale development in a sample of 812 undergraduate students. Item reduction analysis of the original 75 items used to develop the 52-item MACE produced a 58-item Chinese version.” What is the “52-item MACE” that is referred to here? Either add information on what this version is (e.g. it is the standard US version), or remove the reference to this unidentified version. I would suggest the latter.

Thanks for your suggestion. We have removed the reference to the unidentified version as follows.

“Item reduction analysis of the original 75 items produced a 58-item Chinese version with ten subdimensions.” (p. 2)

2. The authors write: “The new scale showed the same factor structure as the original MACE”. It is not clear to me what they mean here, since they did not perform any factor analysis to extract subdimensions on the MACE. Instead, the authors relied on the 10 existing dimensions of MACE and used Rasch analysis on each of the dimensions to reduce number of items. Indeed, in Methods they write “We aimed to develop ten subscales for the Chinese version of the MACE, corresponding to the ten types of maltreatment assessed by the MACE-X.”. Hence, it appears that the reference to factor structure of the Chinese MACE in the abstract, should be removed or reformulated.

Thanks for pointing this. To avoid confusion, we removed this sentence.

3. At the end of the introduction, the authors should specify which psychometric analysis they will perform: test-retest, convergent validity, discriminant validity etc.

True. We specified this at the end of the introduction as follows.

“Rasch modeling, test-retest reliability, convergent validity and predictive validity would be examined.” (p. 5)

4. Methods section may start out with a paragraph that describes the study design.

Thanks for pointing this. We described the study design and procedures to the beginning of the ‘Method’ section in revised manuscript. (p. 6)

5. Part 2.1 on participants: At the end of the paragraph, the authors write that “109 completed all measures again three weeks later to assess test-retest reliability”. I had thought that it was only MACE-CH that was administered again after three weeks, with the other instruments administered only at baseline?

True. Only the Chinese version of MACE-X was completed again after three weeks. We have modified the description about this. (p. 7)

6. At the end of the Results section, under the heading “Recollected time of exposure”, the authors present some data on age of exposure. I could not find any notice or description in Methods that such an analysis should be performed or how it can be understood. In results, some aspects of the methods are noted, but this should be moved to Methods. Moreover, it is not fully clear to me (it does not come clearly across) what the authors actually did, and which results they found in terms of ages of exposure – which they present in table 16. Did they find that, for each subdimension, certain ages occurred more often than others?

Thanks for pointing this. This part is not important and seems redundant. To avoid confusion, we deleted this paragraph in revised manuscript. 

Reviewer #2

1.Please add a comment if you plan making the MACE-CH fully available (open access, upon request)

Thanks for pointing this. The MACE-CH will be provided as open access. We added this point in the ‘Conclusion’ section as follows.

“The MACE-CH will be provided as open access to help facilitate its use.” (p. 22)

2. The MACE-risk associations are not limited to psychopathology (“identify periods of development when childhood maltreatment maximally increases vulnerability to psychopathology“). Health outcomes?

True. We modified this description in revised manuscript as follows.

“identify periods of development when childhood maltreatment maximally increases vulnerability to psychopathology and health outcomes” (p. 2)

3. MACE can also be used in clinical settings and is not only limited to research.

True. We modified this description in revised manuscript as follows.

“This new instrument will be a valuable tool for Chinese researchers and psychiatrist to ascertain the type and timing of exposure to maltreatment.” (p. 2)

4. when you refer to the population-attributable risk. Do you mean the population-attributional risk fraction?

Yes. We replaced ‘population-attributable risk’ with ‘population-attributional risk fraction’ in revised manuscript. (p. 3)

5. Methods: please specify all important parameters that were set e.g. for random effects in linear mixed effect models.

Thanks for pointing this. After careful discussion, we deleted the paragraph describing the linear mixed effect models. Thus, none will be changed here.

6. Please report how you have made sure, that participants reported events correctly (e.g., an event that happened at the age of “5”/”0” needs to be checked for the 6th /first year of life)….

Thank you. During data collection, we gave subjects the instructions about how to report events correctly. For example, from birth to age one (12 months), they were instructed to tick ‘1’.

7. Please specify, which items were considered for potential inclusion of each subscales.

True. These items have been shown in Table 1 to Table 10.

8. How was the subsample selected that did the retest of the MACE-CH?

During data collection at baseline, we asked participants whether they could complete the MACE-X again after three months. If they agree, we will invite them to take the retest. We added this information into the ‘Procedure’ section in revised manuscript. (p. 6)

9. Please explain why Rasch models cannot export sufficient person parameters? And why subscale with 4 items were not scored as 0 = 0, 1= 2.5, 2=5, 3 = 7.5 and 4 = 10?

Based on the number of items positively endorsed by participants and person parameters in the Rasch model, subscales that included 5 or more items were scored for severity of exposure to the latent category. Typically these mean-centered logit scores range from -4 to +4. For the purposes of the current study they were further recalibrated to range from 0–10, enabling total exposure severity levels across the 10 subscales to fall in the range of 0 to 100. In this process, subscales with 4 items could not get mean-centered logit scores due to insufficient item parameters. 

We believe that when subscales with 4 items were scored as ‘0 = 0, 1= 2.5, 2=5, 3 = 7.5 and 4 = 10’, the results are similar. In current study, subscales that contained 4 items were scored 0, 3, 5, 8 and 10. This scoring algorithm is aligned with that in the standard US version of MACE.

10. The age range of the present sample is restricted 18-26years. Would also another slitting would be interesting to detect differential responding for women and men?

True. We have added the results of Andersen's likelihood ratio tests regarding for women and men in revised manuscript.

For instance:

“the Andersen test was also not significant when splitting for gender, χ2(4) = 1.822, p = .769.”

11. The authors seem to consider more strict ranges of infits and outfits for the items as compared to description in the methods (1.3 and 0.7 instead of 1.5 and 0.5). Please be consistent.

Thanks for pointing this. We have modified the description in revised manuscript.

12. I would recommend not to mention the scaled scores when reporting the number of positively endorsed items for cut off severity (Parental verbal abuse).

Thanks for this suggestion. We didn’t mention the scaled scores when reporting the number of positively endorsed items for cut off severity in revised manuscript.

13. Non-verbal emotional abuse: when “splitting”… instead of “slipping”.

Thanks. We have corrected this error.

14. Sexual abuse: The df for the Andersen test was 6. Please report which item was dropped for this analysis. And please check the df for other reported tests (e.g., witnessed violence to siblings).

Thanks for pointing this. There is a clerical error and we have corrected it. We also checked the df for other results.

15. Please also report test-retest reliability of the MACE duration (years with a multiplicity score ≥ 1 (ranging from 0 to 18).

True. We further reported test-retest reliability of the MACE duration (r = 0.812, p < 10-16) in revised manuscript. (p. 17)

16. There is an imbalance in number of items per perpetrator. Only two items of parental sexual abuse were included, while for other adults 4 items were included. This means, that two potentially important items were not scored if endorsed (1) parents attempted to have any type of sexual intercourse with you (2) parents had sexual intercourse with you. From the item difficulties it appears to be rarely reported events. Please revise the subscale or discuss this issue. Would you expect different results when assessing these items in highly exposed individuals (e.g. individuals with mental illness)?

True. These two items you mentioned appear to be rarely reported events, contributing to very low Outfit MSQ ( < 0.2 ). Thus, they are not included in our sexual abuse subscale. We expect different results when assessing these items in clinical populations. This point has been mentioned as a limitation in revised manuscript as follow.

“There is an imbalance in number of items per perpetrator within sexual abuse subscale. Only two items of parental sexual abuse were included, while for other adults 4 items were included. Other items of parental sexual abuse appear to be rarely reported events, contributing to very low Outfit MSQ. Thus, they are not included in final subscale of sexual abuse. It would be valuable in future to test whether there are different results when these items of parental sexual abuse are assessed in highly exposed individuals (e.g. individuals with mental illness).” (p. 21)

17. Table 12: Please specify the parameter of the MACE that you have used for this correlation.

Table 12 showed the test-retest reliability for MACE total scores at each age from years 1 to 18. We specified this in revised manuscript. (p. 17)

Reviewer #3

1. The only thing worth correcting would be in the second last paragraph, line 3 of the discussion section, where the Khan et al paper should be referenced to 2015 and not 2017.

Sorry for this. We have corrected this error in revised manuscript.

2. There were only two minor comments, for future considerations. The duration for testing the test-retest reliability which you have acknowledged as a limitation, was not ideal. Additionally, I wish you had used a scale or two in addition to the Depression and Anxiety Stress Scales.

Thanks for pointing this. We added these two points as limitations in revised manuscript as follows.

“Third, the duration for testing the test-retest reliability is not ideal. Future work using a longer duration would be better. Last, in addition to the Depression and Anxiety Stress Scales, more scales rating psychopathology should be used to test the predictive power of MACE-CH.” (p. 22)

We are very grateful to you for the careful and thoughtful feedback you have provided us. Regardless of the final status of our paper, we are confident that it is much better, thanks to your efforts on our behalf.

---

## [Decision Letter · Decision Letter 1]

10 Jun 2022

PONE-D-22-05087R1The Chinese version of the Maltreatment and Abuse Chronology of Exposure (MACE) scale: Psychometric properties in a sample of young adultsPLOS ONE

Dear Dr. Zhu,

Thank you for submitting your manuscript to PLOS ONE. After careful consideration, we feel that it has merit but does not fully meet PLOS ONE’s publication criteria as it currently stands. Therefore, we invite you to submit a revised version of the manuscript that addresses the points raised during the review process.

 All three reviewers expressed their enthusiasm for your manuscript. Before accepting your work, I would appreciate if you could address a few minor comments raised by reviewer 1. Based on your response I believe I can make then a final editorial decision.

We look forward to receiving your revised manuscript.

Kind regards,

Torsten Klengel, MD PhD

Academic Editor

PLOS ONE

Journal Requirements:

Reviewers' comments:

Reviewer's Responses to Questions

**Comments to the Author**

1. If the authors have adequately addressed your comments raised in a previous round of review and you feel that this manuscript is now acceptable for publication, you may indicate that here to bypass the “Comments to the Author” section, enter your conflict of interest statement in the “Confidential to Editor” section, and submit your "Accept" recommendation.

Reviewer #1: (No Response)

Reviewer #2: All comments have been addressed

Reviewer #3: All comments have been addressed

2. Is the manuscript technically sound, and do the data support the conclusions?

Reviewer #1: Yes

Reviewer #2: Yes

Reviewer #3: Yes

3. Has the statistical analysis been performed appropriately and rigorously? 

Reviewer #1: Yes

Reviewer #2: Yes

Reviewer #3: Yes

4. Have the authors made all data underlying the findings in their manuscript fully available?

Reviewer #1: (No Response)

Reviewer #2: Yes

Reviewer #3: Yes

5. Is the manuscript presented in an intelligible fashion and written in standard English?

Reviewer #1: Yes

Reviewer #2: Yes

Reviewer #3: Yes

6. Review Comments to the Author

Reviewer #1: Prior comments have been adequately responded to.

Below are two further comments that need to be adressed, regarding the DAAS (Methods) and the comparison between MACE multiplicity scores and QACE-R with respect to depression (Results). I have also added some minor comments on language/ grammer.

Abstract – last sentence, change “psychiatrist” to “psychiatrists”

Introduction

Page 4, paragraph starting with “Although the..”. Sentence in midst of paragraph, change “in” to “of”, so that sentence reads “This consideration is critically important as there may be periods of development when exposure to specific types of maltreatment is maximally associated with vulnerability for psychopathology and alterations in the structure and function OF stress-susceptible brain regions.”

Page 5, paragraph starting with “The MACE has shown”, second sentence: Rewrite to something like - “The authors of the MACE have also made available the original set of 75 items, called the MACE-X, from which THEY EXTRACTED A 52 ITEM US VERSION OF THE MACE”.

Page 5, same paragraph as above. In the second last sentence, it is not necessary to once more state what MACE stands for (Maltreatment and…), since this has been done earlier in the paper. Hence, change the sentence to “Thus, this study aimed to (1) establish the MACE-CH, the Chinese version of the MACE scale,..”

Methods

Please add a paragraph in Methods that succinctly describes items/ content, item scaling, anxiety and depression subscales, and psychometric properties of the Depression Anxiety Stress Scales (DASS) scale (this seems to be lacking).

Under the heading “Initial items”, the content of the first sentence has been mentioned before, so this first sentence can be removed/ deleted. Likewise with the second sentence, it can also be removed. Third sentence can then begin the paragraph,with wordings changed to something like “The 75 item MACE-X assesses ten..”

Under the heading “Evaluation for item inclusion in a subscale”, second paragraph third sentence beginning with “Infit depicts” – the word “that” is repeated twice, remove one of them.

Sentence that continues from page 10 to page 11, second part of the sentence: it appears a word or two are missing after the word “the”: “First, ordinary least squares regression was performed to calculate .., using the to ascertain whether MACE had significantly stronger or weaker predictive power than the comparator scale (i.e., QACE-R or CTQ).”

Results

Page 16, subscale for emotional neglect. First sentence, change “the night” to “the nine”.

In the last sentence in Results, page 19, regarding results from multiple regression, the authors write: “Further, variance decomposition analyses found that MACE Severity and Multiplicity scores explained more variance in symptom ratings of anxiety and depression than the CTQ and QACE-R scores did (Table 14 and Table 15).” However, when inspecting the multiple regression parts and variance decomposition parts in Table 15, it does not seem to me that the MACE Multiplicity scores explained more variance than the QACE-R with respect to depressive symptoms: MACE Multi - Beta = 0.100 /p = 0.134 / 2.93% vs QACE-R - Beta = 0.192 / p = 0.004 and 4.17%. Please clarify.

Reviewer #2: All my comments have been addressed carefully.

You may want to consider as well psychologists (beside psychiatrists) in the following sentences:

“This new instrument will be a valuable tool for Chinese researchers and psychiatrist to ascertain the type and timing of exposure to maltreatment.” (p. 2)

Reviewer #3: Dear Authors,

The manuscript was well-written to begin with. With your revisions, its even better now.

Congratulations.

7. PLOS authors have the option to publish the peer review history of their article (what does this mean?). If published, this will include your full peer review and any attached files.

Reviewer #1: **Yes: **Roar Fosse

Reviewer #2: **Yes: **Inga Schalinski

Reviewer #3: **Yes: **Alaptagin Khan

---

## [Author Response · Author response to Decision Letter 1]

13 Jun 2022

Jun 11, 2022

Torsten Klengel, MD PhD

Academic Editor

PLOS ONE

Manuscript Number: PONE-D-22-05087R1

Dear Dr. Klengel:

Thank you for the constructive feedback from you and the reviewers of our manuscript, “The Chinese version of the Maltreatment and Abuse Chronology of Exposure (MACE) scale: Psychometric properties in a sample of young adults”, Manuscript Number: PONE-D-22-05087R1. We have revised the manuscript and incorporated all suggestions and comments from reviewers. Below are our responses in detail.

Reviewer #1

Abstract – last sentence, change “psychiatrist” to “psychiatrists”

Thank you. We have changed “psychiatrist” to “psychiatrists”. (p. 2)

Introduction

Page 4, paragraph starting with “Although the..”. Sentence in midst of paragraph, change “in” to “of”, so that sentence reads “This consideration is critically important as there may be periods of development when exposure to specific types of maltreatment is maximally associated with vulnerability for psychopathology and alterations in the structure and function OF stress-susceptible brain regions.”

Thank you. We have changed ‘in’ to ‘of’. (p. 4)

Page 5, paragraph starting with “The MACE has shown”, second sentence: Rewrite to something like - “The authors of the MACE have also made available the original set of 75 items, called the MACE-X, from which THEY EXTRACTED A 52 ITEM US VERSION OF THE MACE”.

Thank you. We have modified this sentence according to your advice. (p. 5)

Page 5, same paragraph as above. In the second last sentence, it is not necessary to once more state what MACE stands for (Maltreatment and…), since this has been done earlier in the paper. Hence, change the sentence to “Thus, this study aimed to (1) establish the MACE-CH, the Chinese version of the MACE scale,..”

Thank you. We have modified this sentence according to your advice. (p. 5)

Methods

Please add a paragraph in Methods that succinctly describes items/ content, item scaling, anxiety and depression subscales, and psychometric properties of the Depression Anxiety Stress Scales (DASS) scale (this seems to be lacking).

True. We provided a succinct description for DASS in revised manuscript as follows. (p. 10)

‘This measure has shown good reliability and validity in prior research [35]. For the purposes of this study, we used the anxiety and depression subscales as the outcome variables. Participants were instructed to assess their anxiety symptoms (e.g., “I was aware of dryness of my mouth”) or depressive symptoms (e.g., “I found it difficult to work up the initiative to do things.”) during the past week. Fourteen items (seven for each) were rated on a four-point scale ranging from 1 (did not apply to me at all) to 4 (applied to me very much of the time). The responses were averaged across items, with higher scores indicating higher anxiety or depressive symptoms.’

Under the heading “Initial items”, the content of the first sentence has been mentioned before, so this first sentence can be removed/ deleted. Likewise with the second sentence, it can also be removed. Third sentence can then begin the paragraph,with wordings changed to something like “The 75 item MACE-X assesses ten..”

True. We deleted these two sentences in revised manuscript. (p. 7)

Under the heading “Evaluation for item inclusion in a subscale”, second paragraph third sentence beginning with “Infit depicts” – the word “that” is repeated twice, remove one of them.

True. We have deleted the redundant term ‘that’. (p. 8)

Sentence that continues from page 10 to page 11, second part of the sentence: it appears a word or two are missing after the word “the”: “First, ordinary least squares regression was performed to calculate .., using the to ascertain whether MACE had significantly stronger or weaker predictive power than the comparator scale (i.e., QACE-R or CTQ).”

True. We have modified this sentence in revised manuscript. (p. 11)

Results

Page 16, subscale for emotional neglect. First sentence, change “the night” to “the nine”.

Thanks. We have changed ‘night’ to ‘nine’. (p. 16)

In the last sentence in Results, page 19, regarding results from multiple regression, the authors write: “Further, variance decomposition analyses found that MACE Severity and Multiplicity scores explained more variance in symptom ratings of anxiety and depression than the CTQ and QACE-R scores did (Table 14 and Table 15).” However, when inspecting the multiple regression parts and variance decomposition parts in Table 15, it does not seem to me that the MACE Multiplicity scores explained more variance than the QACE-R with respect to depressive symptoms: MACE Multi - Beta = 0.100 /p = 0.134 / 2.93% vs QACE-R - Beta = 0.192 / p = 0.004 and 4.17%. Please clarify.

True. To avoid confusion, we modified this sentence as follows.

“Further, variance decomposition analyses found that MACE Severity scores explained more variance in symptom ratings of both anxiety and depression than the CTQ scores did (Table 14), and MACE Multiplicity scores explained more variance in symptom ratings of anxiety than the QACE-R scores did (Table 15).” (p. 19)

We are very grateful to you for the careful and thoughtful feedback you have provided us. Regardless of the final status of our paper, we are confident that it is much better, thanks to your efforts on our behalf.

---

## [Editor Report · Decision Letter 2]

16 Jun 2022

The Chinese version of the Maltreatment and Abuse Chronology of Exposure (MACE) scale: Psychometric properties in a sample of young adults

PONE-D-22-05087R2

Dear Dr. Zhu,

We’re pleased to inform you that your manuscript has been judged scientifically suitable for publication and will be formally accepted for publication once it meets all outstanding technical requirements.

Kind regards,

Torsten Klengel, MD PhD

Academic Editor

PLOS ONE
---

## [Editor Report · Acceptance letter]

22 Jun 2022

PONE-D-22-05087R2 

The Chinese version of the Maltreatment and Abuse Chronology of Exposure (MACE) scale: Psychometric properties in a sample of young adults 

Dear Dr. Zhu:

I'm pleased to inform you that your manuscript has been deemed suitable for publication in PLOS ONE. Congratulations! Your manuscript is now with our production department. 

Kind regards, 

on behalf of

Dr. Torsten Klengel 

Academic Editor

PLOS ONE